# Differential Ability of Spike Protein of SARS-CoV-2 Variants to Downregulate ACE2

**DOI:** 10.3390/ijms25021353

**Published:** 2024-01-22

**Authors:** Yosuke Maeda, Mako Toyoda, Takeo Kuwata, Hiromi Terasawa, Umiru Tokugawa, Kazuaki Monde, Tomohiro Sawa, Takamasa Ueno, Shuzo Matsushita

**Affiliations:** 1Department of Microbiology, Faculty of Life Sciences, Kumamoto University, Kumamoto 860-8556, Japanmonde@kumamoto-u.ac.jp (K.M.); sawat@kumamoto-u.ac.jp (T.S.); 2Joint Research Center for Human Retrovirus Infection, Kumamoto University, Kumamoto 860-0811, Japan; makotoyo@kumamoto-u.ac.jp (M.T.); tkuwata@kumamoto-u.ac.jp (T.K.); uenotaka@kumamoto-u.ac.jp (T.U.); shuzo@kumamoto-u.ac.jp (S.M.)

**Keywords:** SARS-CoV-2, COVID-19, ACE2, downregulation, S protein, VOCs

## Abstract

Severe acute respiratory syndrome coronavirus 2 (SARS-CoV-2) is the causative agent of coronavirus disease 19 (COVID-19) and employs angiotensin-converting enzyme 2 (ACE2) as the receptor. Although the expression of ACE2 is crucial for cellular entry, we found that the interaction between ACE2 and the Spike (S) protein in the same cells led to its downregulation through degradation in the lysosomal compartment via the endocytic pathway. Interestingly, the ability of the S protein from previous variants of concern (VOCs) to downregulate ACE2 was variant-dependent and correlated with disease severity. The S protein from the Omicron variant, associated with milder disease, exhibited a lower capacity to downregulate ACE2 than that of the Delta variant, which is linked to a higher risk of hospitalization. Chimeric studies between the S proteins from the Delta and Omicron variants revealed that both the receptor-binding domain (RBD) and the S2 subunit played crucial roles in the reduced ACE2 downregulation activity observed in the Omicron variant. In contrast, three mutations (L452R/P681R/D950N) located in the RBD, S1/S2 cleavage site, and HR1 domain were identified as essential for the higher ACE2 downregulation activity observed in the Delta variant compared to that in the other VOCs. Our results suggested that dysregulation of the renin–angiotensin system due to the ACE2 downregulation activity of the S protein of SARS-CoV-2 may play a key role in the pathogenesis of COVID-19.

## 1. Introduction

Severe acute respiratory syndrome coronavirus 2 (SARS-CoV-2) is the causative pathogen of coronavirus disease 19 (COVID-19), a disease that can lead to life-threatening conditions such as pneumonia [1]. It was first identified during an outbreak in December 2019 in Wuhan, China, and has rapidly spread worldwide [2] The World Health Organization (WHO) declared COVID-19 a pandemic in March 2020. As of October 2023, there had been more than 270 million confirmed cases of COVID-19, with more than 2 million deaths reported by the WHO [3].

Similar to other RNA viruses, SARS-CoV-2 undergoes changes that affect its properties over time. Among these changes, variants that can influence transmissibility, disease severity, vaccine effectiveness, diagnostic tools, and therapeutic medicines are considered as variants of concern (VOCs) by the WHO [4]. Previous VOCs were Alpha (B.1.1.7), Beta (B1.351), Gamma (P.1), Delta (B.1.617.2), and Omicron (B.1.1.529) [5]. These variants demonstrated increased transmissibility and reduced susceptibility to neutralization compared to the wild-type Wuhan strain. However, the pathogenic properties of the VOCs vary. For instance, the delta variant is associated with severe illness, whereas the omicron variant is less pathogenic [6,7,8,9]. Most of the mutations were accumulated in the S gene of these VOCs, which allows for viral attachment and entry into host cells through interactions with its receptor, angiotensin-converting enzyme 2 (ACE2) [10,11,12,13,14].

However, the viral properties of the S protein that affect the pathogenicity in these variants are still to be determined. Additionally, the SARS-CoV, identified in 2003 as the etiological agent of severe acute respiratory syndrome (SARS), also employs ACE2 as the receptor. It has been shown that the S protein of SARS-CoV downregulates ACE2, a key molecule in the renin–angiotensin system (RAS) [15,16,17]. As the RAS plays crucial roles not only in the regulation of circulation, such as blood pressure and fluid balance, but also in the regulation of inflammation, RAS dysregulation may induce systemic inflammation in COVID-19 [18,19,20,21,22]. Indeed, the downregulation of ACE2 by the S protein of SARS-CoV induces lung injury in mice [16]. Furthermore, the ACE2 downregulation activity of SARS-CoV was higher than that of human coronavirus—NL63—which also employs ACE2 as the receptor, but it is less pathogenic and only causes the common cold in humans [17]. Therefore, in this study, we analyzed the ability of SARS-CoV-2 to downregulate ACE2 expression. The S protein of SARS-CoV-2 was found to downregulate ACE2, and this downregulation was correlated with the pathogenic properties of the VOCs.

## 2. Results

### 2.1. The S Protein of SARS-CoV-2 Downregulates ACE2

The expression of ACE2 in target cells is essential for the cellular entry of SARS-CoV-2. However, it is possible that ACE2 expression in virus-infected cells could lead to superinfection, produce fewer infectious virions, or inhibit viral release due to the interaction of the S protein in the endoplasmic reticulum (ER), Golgi, or plasma membrane with ACE2, as previously reported in different enveloped viruses [23,24,25,26,27,28,29,30,31]. Therefore, the downregulation of ACE2 in the infected cells occurs through certain mechanisms (s). Given that ACE2, which also serves as the receptor for SARS-CoV, is downregulated in SARS-CoV infection by its S protein, as previously described [16], we aimed to determine whether ACE2 is similarly downregulated by the S protein of SARS-CoV-2. To this end, we established ACE2-expressing 293T cells (293T/ACE2) and assessed ACE2 expression levels in the presence of S protein by Western blotting. We found a reduction in ACE2 expression upon the introduction of the S protein, whereas the S protein level in ACE2-expressing cells remained relatively constant (Figure 1A). We further examined whether fluorescent protein-tagged ACE2 (ACE2-Venus) was downregulated by the S protein using flow cytometry. In the S-protein-positive cell population, the level of ACE2-Venus was markedly diminished. In contrast to the Western blot, the surface expression level of the S protein was also reduced in the presence of ACE2 (Figure 1B). These results confirmed that ACE2 is downregulated by the S protein of SARS-CoV-2.

### 2.2. The Expression of ACE2 Impeded the Infectivity of Pseudovirus with the S Protein of SARS-CoV-2

To evaluate the impact of ACE2 expression in infected cells on the infectivity of nascent progeny virions, we produced an HIV-based pseudotype SARS-CoV-2 S in the presence or absence of FLAG-tagged ACE2 and assessed its infectivity in a single round of infection. Given the relatively low infectivity of the pseudovirus with the wild-type (Wuhan strain) S protein of SARS-CoV-2, we used the D614G S protein for the pseudovirus production and infection of ACE2-expressing 293T cells [32,33]. ACE2 expression in pseudovirus-producing cells led to an approximately 50-fold reduction in the infectivity of the pseudovirus with the S protein, whereas the pseudovirus with VSVG showed no change in infectivity in the presence of ACE2 (Figure 2A). Western blot analysis demonstrated that ACE2 expression in pseudovirus-producing cells was diminished in the presence of the S protein (Figure 2B), and the expression level of the S protein remained unaffected by ACE2 expression. However, the incorporation of the S protein into pseudoviruses was less efficient in the presence of ACE2 in producer cells, whereas the incorporation of VSVG was comparable in the presence and absence of ACE2, confirming that the incorporation of the S protein into pseudoviruses is crucial for maintaining their infectivity.

To investigate whether higher levels of ACE2 in virus-infected cells affect the infectivity of authentic SARS-CoV-2, we established a clone expressing higher amounts of ACE2 (Figure 2C) to confirm whether ACE2 expression in virus-producing cells affects its infectivity. Our results demonstrated that the replication kinetics of authentic SARS-CoV-2 in the high-ACE2-expressing clone was nearly comparable to those of the parental cells (Figure 2D), suggesting that ACE2 downregulation in virus-producing cells occurs during authentic SARS-CoV-2 infection.

### 2.3. Interaction of ACE2 with the S Protein of SARS-CoV-2 in the Cytoplasmic Compartment of the Cells

To verify the interaction between the S protein and ACE2 within cells, we co-transfected FLAG-tagged ACE2 and the S protein into 293T cells. Subsequently, ACE2 or the S protein was immunoprecipitated using anti-FLAG or a monoclonal antibody recognizing the N-terminal domain (NTD) of the S protein, respectively. Our results demonstrated the successful immunoprecipitation of the S protein with anti-FLAG or ACE2-FLAG with an anti-S antibody (Figure 3A). Furthermore, confocal laser microscopy revealed that ACE2 colocalized with the S protein in the cytoplasmic compartment, exhibiting a speckled pattern (Figure 3B). These findings confirm a direct interaction between the S protein and ACE2 in the cytoplasm of the same cells, suggesting a potential mechanism for ACE2 downregulation.

### 2.4. The S Protein Degrades ACE2 in the Lysosomal Compartment through the Endocytic Pathway

To investigate the regulatory mechanism of ACE2 via the S protein, we examined whether ACE2 expression could be restored using inhibitors targeting various protein degradation pathways, including endocytic, autophagic, and proteasomal pathways. Our findings revealed that inhibitors of endosome acidification, such as ammonium chloride, chloroquine, and bafilomycin A1 (BFLA1), effectively restored ACE2 expression, as observed by Western blot analysis. In contrast, the autophagy inhibitor 3-methyl adenine (3MA) and proteasome inhibitor MG132 did not have a significant effect (Figure 4A).

Additionally, a protease inhibitor cocktail targeting lysosomal proteases partially restored ACE2 expression in the presence of the S protein (Figure 4A). Furthermore, ACE2 co-localized with the late endosome or lysosome marker Rab9a in a punctate pattern when the S protein was present, whereas ACE2 showed a homogenous distribution in the cytoplasm in the absence of the S protein (Figure 4B). These results suggest that ACE2 translocates from the endosomal to the lysosomal compartment and is degraded by lysosomal proteases.

### 2.5. Cytoplasmic Domains of ACE2 and S Protein Are Not Essential for the Downregulation of ACE2

As ACE2 degradation was suggested to occur via the endocytic pathway, it was hypothesized that the cytoplasmic domain of ACE2 mediated its degradation of ACE2 through interactions with the endocytic sorting motif (YXXΦ) present in ACE2 (Appendix A), which binds the μ2 subunit of AP-2 adaptors [34]. Therefore, it is plausible that this motif plays a role in ACE2 downregulation. To investigate this hypothesis, we generated several FLAG-tagged ACE2 mutants lacking the cytoplasmic domain (Appendix A) and introduced them to the 293T cells in the presence or absence of the S protein.

Surprisingly, all cytoplasmic-domain-lacking ACE2 mutants were downregulated, similar to wild-type ACE2, using Western blot analysis, indicating that the cytoplasmic domain of ACE2 was not necessary for ACE2 downregulation. Similarly, the S protein lacking the cytoplasmic domain also mediated ACE2 downregulation (Appendix A), suggesting that the cytoplasmic domain of the S protein was also dispensable in the downregulation of ACE2.

### 2.6. Differential Ability of S Protein from SARS-CoV-2 Variants to Downregulate ACE2

SARS-CoV infection or the injection of recombinant SARS-CoV S protein reduced ACE2 expression and induced lung injury in a mouse model [16]. Additionally, the downregulation of ACE2 in COVID-19 has been hypothesized to be associated with its pathogenesis [35]. Therefore, we investigated whether the S proteins from previous VOCs may have different abilities to downregulate ACE2. For instance, Omicron (B.1.1.527) parent lineage infections, such as BA.1 or BA.2, are associated with a lower disease severity than the Delta (B.1.617.2) variant [6,33].

To explore this, we introduced the S protein of previous VOCs into ACE2-expressing 293T cells and examined the ACE2 expression levels. We found that most VOCs, including Alpha, Beta, Gamma, and Delta variants, had a comparable ability to downregulate ACE2, whereas a previously circulating Omicron variant, BA.1, showed a significantly lower downregulation ability in both Western blot (Figure 5A) and flow cytometric analyses (Figure 5B). To quantitatively analyze the ability of ACE2 to downregulate the S protein in different SARS-CoV-2 variants, we developed a novel assay using nanoluciferase-linked ACE2 (ACE2-nLuc).

We transduced ACE2-nLuc into 293T cells using a lentiviral vector and selected a clone expressing the highest level of ACE2-nLuc. ACE2 regulation by the S protein was determined by the reduction in nanoluciferase activity in cells transfected with the S protein compared to that in cells transfected with an empty vector. Using this system, we confirmed that the Omicron variant had a significantly lower downregulation activity than other VOCs, including the Delta variant (Figure 5C). The Gamma variant showed relatively lower downregulation activity compared to the Delta variant in this assay, but this was not consistently observed in Western blot analysis.

### 2.7. Determination of the Reduced ACE2 Downregulation Ability in Omicron Variants via the Receptor-Binding Domain and Heptad Repeat Domains of S the Protein

To identify the regions responsible for the reduced ACE2-downregulation ability of the Omicron variant, we examined the S protein sequence, which revealed multiple mutations in the NTD, receptor binding domain (RBD), the S1/S2 cleavage site, and the S2 subunit, including the HR1 domain (Figure 6A). Consequently, we generated S protein chimeras by combining sequences from Omicron and Delta variants. This was achieved using shared restriction sites (*Hind III*, *KflI*, *BsrGI*, and *XbaI*), and the 5′ *HindIII-KflI* fragment was further divided into two domains, NTD and RBD, through standard molecular cloning (Figure 6B). The middle *KflI-BsrGI* fragment contained the furin S1/S2 cleavage site, whereas the 3′ *BsrGI-XbaI* fragment contained the S2′ cleavage site and S2 subunit, including the fusion peptide (FP) and HR1/HR2 domains.

Thereafter, we assessed the ACE2 downregulatory activity of each chimeric S protein. The chimera with the RBD from Omicron in the Delta variant background (dodd) exhibited a lower ACE2 downregulation activity than the Delta variant (Table 1). Additionally, the inclusion of the 3′ fragment from the Omicron (dodo), resulting in a similar ACE2 downregulation activity to that of the Omicron variant, confirmed the role of RBD and the 3′ fragment, including the S2′ cleavage site, FP, and HR1/HR2 domains of Omicron, in the reduced downregulation activity of ACE2 (Table 1).

We assessed the fusion activity of each S protein, considering that the Omicron variant exhibited lower fusion activity than the wild-type Wuhan strain, whereas the delta variant had higher fusion activity. It is plausible that the fusion mediated by the S protein of each variant leads to the downregulation of ACE2. Therefore, the fusion activity of the S protein was analyzed using a dual-split protein (DSP)-based cell–cell fusion assay, as previously described [36,37].

We observed that the S protein of the Delta variant exhibited the highest fusion activity, whereas the S protein of the Omicron variant demonstrated the lowest fusion activity. These results indicated that the fusion activity of the S protein primarily determines the ACE2 downregulation activity. However, in contrast to the ACE2 downregulation activity, substitutions with a single fragment other than the NTD from the Omicron in the Delta variant background (dodd, ddod, dddo) led to a similar lower fusion activity compared to that of the Delta variant (Table 1). Substitutions with the NTD from the Delta variant in the Omicron background (dooo) achieved the lowest fusion activity, similar to that of the Omicron variant, whereas the inclusion of the RBD from the Delta variant (ddoo) did not significantly increase the fusion activity (Table 1). This suggests that major roles are played by the S1/S2, S2′ cleavage sites, FP, and HR1/HR2 domains, but not the NTD/RBD, in the lower fusion activity of the Omicron variant. Altogether, the regions responsible for the ACE2 downregulation activity and the fusion activity of the S protein somewhat overlapped, but were not simply correlated.

### 2.8. The Determination of the Increased ACE2 Downregulation Ability in the Delta Variants through the Combination of Three Substitutions in L452R/P681R/D950N of the S Protein

Next, we aimed to identify the amino acids responsible for the high ACE2 downregulatory activity of the Delta variant. Since NTD has not been reported to be associated with ACE2 binding, fusion activity, and pathogenicity, and looking at our chimeric data between the Omicron and Delta variants, we focused on the RBD in the S1 subunit, S1/S2 cleavage site, and S2 subunit of the S protein from the Delta variant.

In these domains, the Delta variant had five mutations, namely L452R, T478K, D614G, P681R, and D910N, compared to the Wuhan strain (Figure 6A). Among them, we further investigated L452R, P681R, and D950N, because the L452R mutation increases binding affinity to ACE2 [38], P681R, and D950N, which enhance the fusion activity [39], while P681R has been reported to be involved in pathogenicity [40].

Our findings revealed that the single-mutation P681R induced a slightly higher ACE2 downregulation than the other single mutations. The double-mutation P681R/D950N enhanced ACE2 downregulation to some extent (Table 2). However, all three combined mutations L452R/P681R/D950N were necessary for the ACE2 downregulation activity, which was comparable to that of the Delta variant (Table 2).

In contrast, a single P681R mutation significantly enhanced the fusion activity, whereas single mutations in L452R or D950N did not alter the fusion activity, as previously reported [39] (Table 2). This confirmed that P681R is the principal mutation responsible for the enhanced fusion activity of the Delta variant. The addition of D950N, combined with P681R, further enhanced the fusion activity until it was comparable to that of the Delta variant, emphasizing the importance of P681R and D950N in the higher fusion activity of the Delta variant. These results also suggest that the fusion and ACE2 downregulation activities are not simply correlated.

## 3. Discussion

Similar to SARS-CoV, ACE2 was reported to be the receptor for SARS-CoV-2. The S protein of SARS-CoV-2 interacts with the ACE2 expressed in the target cells to mediate cellular entry [10,11,12,13,14]. However, the receptor molecules infected by the enveloped viruses are generally downregulated to maintain the infectivity of progeny virions. For instance, CD4 molecules in HIV-1-infected cells are downregulated by HIV-1 Env, Nef, and Vpu proteins to avoid superinfection and maintain viral infectivity [23,24,25,29,31]. The receptor molecule for the measles virus, CD150 (SLAM), is also downregulated by its H protein [30]. Similarly, beta coronavirus, SARS-CoV induces ACE2 downregulation [15,16,17]. In this study, the S protein of SARS-CoV-2 also induces ACE2 downregulation at the protein level (Figure 1). The S protein of SARS-CoV-2 has already been reported to downregulate the transcriptional level of ACE2 in primary lung cells [41]. However, our results from co-immunoprecipitation and confocal microscopy experiments indicate that the interaction of the S protein with ACE2 is crucial for the downregulation of ACE2. Notably, the expression of the S protein in the presence of ACE2 was reduced in the flow cytometric analysis compared to the Western blot. This is probably due to the suppression of the cell surface expression of the S protein through the ACE2 expression, resulting in the less efficient incorporation of the SARS-CoV-2 S protein into the pseudovirions (Figure 2B). Experiments using inhibitors targeting several protein degradation pathways, as well as confocal microscopy, have shown that ACE2 is degraded by the S protein in the lysosomal compartment via the endocytic pathway (Figure 4). Reportedly, recombinant SARS-CoV S protein induces ACE2 downregulation by interacting with the cell surface [16]. After the discovery of SARS-CoV-2, the addition of the recombinant SARS-CoV-2 S protein also induced ACE2 downregulation through the endocytosis of ACE2 [42]. In our present study, we found that de novo synthesized S protein and ACE2 also interacted with each other in the same cells, resulting in the degradation of ACE2 in the lysosomal compartment. However, the entry efficiency of the SARS-CoV-2 S pseudovirus produced using ACE2-expressing cells was quite low because of the less efficient incorporation of the S protein into the pseudovirions (Figure 2A,B). In contrast, the authentic SARS-CoV-2 virus has a similar replication fitness in highly expressing ACE2 cells, as shown in our present data (Figure 2D). The coronavirus virion buds into the lumen of the endoplasmic reticulum–Golgi intermediate compartment (ERGIC) and is released from the plasma membrane via the exocytic pathway [43,44,45], whereas most enveloped viruses bud from the plasma membrane via the secretory pathway. However, in the case of HTLV-1 infection, we found that GLUT1, the receptor molecule for HTLV-1, and its Env, were located separately in the same cells to avoid interactions and maintain infectivity [26]. Therefore, it is possible that the authentic SARS-CoV-2 has an alternative pathway for the efficient incorporation of the S protein by changing the compartment for the budding site, in addition to S-protein-mediated ACE2 downregulation.

The differential ACE2 downregulation in the S proteins of SARS-CoV and human coronavirus NL63 has been previously reported [17]. SARS-CoV induces severe lung injury, whereas NL63 causes the common cold, suggesting that the ACE2 downregulation activity of each variant is associated with disease severity. In our present study, we observed that the S protein from majority of the variants exhibited comparable levels of ACE2 downregulation activity, except the Omicron variant, although assay-dependent differences were found among the variants. However, the S protein from the Omicron constantly showed lower ACE2 downregulation activity (Figure 5). Recent clinical data have shown that the recently emerged Omicron variant has lower pathogenicity than the Delta variant [6,7,8,9]. These results suggest that ACE2 downregulation of the S protein of each variant is correlated with pathogenesis. Similarly, our group previously reported that the differential ability of primary HIV-1 Nef to downregulate the HIV-1 receptor CD4 and coreceptor CCR5 is associated with the clinical outcome of HIV-1 infection [46]. This suggests that the downregulation of the enveloped virus receptors is generally associated with its pathogenesis. Although the pathogenic features are dependent on the receptor molecules downregulated by the enveloped viruses, the SARS-CoV-2 receptor ACE2 is a key molecule in the RAS. RAS regulates not only blood pressure, but also multiple functions, such as the modulation of inflammation. ACE2 is expressed in the vascular endothelium, kidneys, and the respiratory epithelium. ACE2 cleaves angiotensin I (AngI) or angiotensin II (AngII) into inactive peptides Ang1-9 or Ang1-7, respectively, thereby maintaining RAS homeostasis [18,19,20]. Therefore, ACE2 downregulation by SARS-CoV-2 induces endothelial dysfunction and inflammation, leading to organ damage such as lung injury [18,19,20,21,22].

In the present study, the Delta variant showed strong activity in downregulating ACE2, while the recent Omicron variant had the lowest activity (Figure 5). These findings correlated well with the fusion activity of these variants, suggesting that the downregulation of ACE2 may be principally attributed to the fusion activity of SARS-CoV-2 variants. However, our chimeric and mutation studies on the S protein of the variants indicated that the regions responsible for ACE2 downregulation and fusion activities were slightly different. For example, the P681R mutation present in the Delta variant was the principal determinant of higher fusion activity, whereas three mutations (P681R/L452R/D950N), but not P681R alone, were necessary for higher ACE2 downregulation activity (Table 2). In our chimeric studies of Delta and Omicron variants, we observed that the RBD and S2 regions were responsible for the lower ACE2 downregulation activity of the Omicron variant (Table 1). However, the RBD from the Omicron variant was dispensable, with lower fusion activity (Table 1). Therefore, ACE2 downregulation is likely caused not only by its fusion activity but also by the binding affinity of the S protein with ACE2, or the proteolytic cleavage efficiency of the S protein through furin [47,48,49] or TMPRSS2 [10,50].

In May 2023, the WHO declared the end of the COVID-19 global health emergency [51]. The number of hospitalized cases decreased, and the case fatality rate became less than 0.3%, which was attributed to the emergence of the Omicron BA.2 subvariant, widespread vaccination, and the development of antivirals [52]. However, concerns remain regarding the emergence of new variants with higher pathogenicity. Therefore, surveillance of newly emerging variants is necessary to prepare for future pandemics. We hope that our strategy for analyzing the pathogenic characteristics of SARS-CoV-2 using ACE2 downregulation will prove helpful in this ongoing effort.

## 4. Materials and Methods

### 4.1. Cell Lines, Reagents, and Viruses

Human embryonic kidney cell line, 293T, and VeroE6/TMPRSS2 were maintained in DMEM (Millipore Sigma, Burlington, MA, USA) supplemented with 10% fetal bovine serum (FBS) (Thermo Fisher Scientific, Waltham, MA, USA) and 1% penicillin/streptomycin (PS) (Nacalai Tesque, Kyoto, Japan). NCI-H292 cells (ATCC CRL-1848) were maintained in RPMI1640 supplemented with 10% FBS and PS.

Chloroquine, NH_4_Cl, 3 methyl adenine (3MA), protease inhibitor cocktail, and MG132 were purchased from Sigma-Aldrich (Burlington, MA, USA). Bafilomycin A1 was purchased from Wako (Kyoto, Japan). A monoclonal antibody against the S1 domain of the SARS-CoV-2 S protein (HL1, Cat. No. GTX635656) and polyclonal rabbit antibody against the RBD region of the S protein (Cat. No. GTX135709) were purchased from GeneTex (Irvine, CA, USA). Anti-ACE2 goat polyclonal antibody (Cat. AF933) was purchased from R&D Systems (Minneapolis, Minnesota, USA). Anti-FLAG (DYLDDDK-tag) mouse mAb was purchased from FujiFilm Wako (Osaka, Japan). Anti-β-actin mouse and rabbit monoclonal antibodies were purchased from Sigma-Aldrich (Burlington, MA, USA) and Cell Signaling Technology (Danvers, MA, USA).

The clinically isolated SARS-CoV-2 lineage and D614G-bearing isolate (B.1.1 lineage, strain TKYE641838; DDBJ ID: LC606020) were provided by the Tokyo Metropolitan Institute of Public Health, Tokyo, Japan. The virus was expanded using VeroE6/TMPRSS2 and the plaque-forming units (PFU) were determined as described previously [38].

### 4.2. Plasmids

The mammalian expression vector with the CAG promoter, pCAGGS-p7, was provided by Dr. Kenzo Tokunaga at the National Institute of Infectious Diseases in Japan. An S expression vector containing the D614G mutation (pC-SARS-S-D614G) was provided by Dr. Kenzo Tokunaga. The SARS-CoV-2 (2019-nCoV) Spike S1 Gene ORF cDNA clone expression plasmid (codon-optimized) was purchased from Sino Biological (Beijing, China), and the VOC expression vectors were constructed as previously described [53]. The human ACE2 gene was cloned into a pCR-TOPO vector (Thermo Fisher Scientific, Waltham, MA, USA) from the cDNA of the NCI-H292 cell line. Expression vectors for ACE2, ACE2-FLAG, and ACE2-Venus were then constructed using the pCAGGS-p7 vector. A retroviral vector for human ACE2 expression was constructed using *HpaI* and *BglII* restriction sites of pMSCV-puromycin (Clontech-TAKARA, Shiga, Japan). A lentiviral vector for the expression of ACE2-nanoluciferase was constructed using pLenti6/V5-D-TOPO (Thermo Fisher Scientific, Waltham, MA, USA). Chimeras between the S protein expression vectors of the Delta and Omicron variants were constructed using shared restriction enzyme sites *HindIII*, *KflI*, *BsrGI*, and *XbaI* in the same vector. Chimeras in the NTD and RBD of the S1 domain between the Delta and Omicron variants were constructed using NEBuilder HiFi DNA assembly (New England Biolabs, Ipswich, MA, USA). The dual-split-reporter vectors, pDSP1-7 and pDSP8-11, were kindly provided by Dr. Jin Gouda of the Institute of Medical Science, The University of Tokyo.

### 4.3. Infection Experiments

The infection with HIV-based SARS-CoV-2 S pseudotyped virus was performed as previously described [26,54,55]. In brief, 293T cells were transfected with pNL-LucΔBglII [54] and the S expression vector with the D614G mutation [33] or VSVG expression vectors, in the absence or the presence of FLAG-tagged ACE2, using Lipofectamine 3000 (Thermo Fisher Scientific, Waltham, MA, USA) according to the manufacture’s protocol. After 48 h of transfection, the recovered viruses were used to infect 293T cells expressing ACE2. After 48 h of infection, the luciferase activity of the infected cells was measured using a One-Glo luciferase assay system (Promega, Madison, WI, USA) and GloMax luminometer (Promega, Madison, WI, USA).

SARS-CoV-2 infection was performed as previously described [38]. In brief, one day prior to infection, 293T cells (10,000 cells) were seeded into a 96-well plate. SARS-CoV-2 (MOI = 0.1) was inoculated at 37 °C for 1 h. The infected cells were then washed, and 180 µL of culture medium was added. The culture supernatant (10 µL) was harvested at the indicated time points and used for real-time RT-PCR to quantify viral RNA copy number (see below).

### 4.4. Real-Time RT-PCR

RT–qPCR was performed as previously described [38]. In brief, 5 μL of culture supernatant was mixed with 5 μL of 2 × RNA lysis buffer (2% Triton X-100 (Nacalai Tesque, Cat# 35501-02), 50 mM KCl, 100 mM Tris-HCl (pH 7.4), 40% glycerol, 0.4 U/μL recombinant RNase inhibitor (Promega, Madison, WI, USA, Cat# N2615)) and incubated at room temperature for 10 min. RNase-free water (90 μL) was added, and the diluted sample (3 μL) was used as the template for real-time RT-PCR, performed according to the manufacturer’s protocol using One Step PrimeScript™ III RT-qPCR Mix (Takara, Kyoto, Japan, Cat# RR600B). N2 (2019-nCoV) (Takara, Kyoto, Japan, Cat# XD0008) was used as the primer and probe. The viral RNA copy number was standardized using a positive-control RNA Mix (2019-nCoV) (Cat# XA0142; Takara, Kyoto, Japan). Fluorescent signals were acquired using the LightCycler 96 system (Roche Diagnostics GmbH, Basel, Switzerland).

### 4.5. Flow Cytometry

The 293T cells were co-transfected with ACE2-Venus (0.3 μg) and S protein (0.2 μg) expression vectors from the VOCs using Lipofectamine 3000 in 24-well plate and recovered after 24 h of culture using enzyme-free cell dissociation buffer (Gibco/Thermo Fisher Scientific, Waltham, MA, USA). The cells were then stained with a rabbit monoclonal antibody against S1 (HL1) or a polyclonal antibody against the RBD of the SARS-CoV-2 S protein (GeneTex, Irvine, CA, USA, Cat No. GTX135709), stained with donkey anti-rabbit IgG conjugated with DyLight^TM^ 649 (BioLegend, San Diego, CA, USA), and fixed with 4% paraformaldehyde. The fixed cells were then acquired using FACSCalibur Flow Cytometer (BD Biosciences, San Jose, CA, USA) and analyzed using FlowJo software version 10.9.0 (FlowJo, LLC, Ashland, OR, USA). The mean fluorescence intensities of the Venus-positive population in the S-protein-positive fraction were calculated from three independent experiments.

### 4.6. Quantitative Measurement of ACE2 Downregulation Activity by the S Protein from VOCs and Its Chimeras

The ACE2-nanoluciferase carrying 293T (293T/ACE2-nLuc) were seeded in 96-well plates. The following day, the S protein expression vectors (50 ng) were transfected using Lipofectamine 3000. Nanoluciferase activity was measured after 48 h of culture using the Nano-Glo luciferase assay system (Promega, Madison, WI, USA) and GloMax luminometer (Promega, Madison, WI, USA). The ACE2-downregulation activity of the S protein from VOCs or chimeras was calculated using the activity of the wild-type S protein as 100%.

### 4.7. Measurement of Fusion Activity of S Protein with ACE2

The fusion activity of the S protein with ACE2 was measured as previously described. In brief, 293T cells were co-transfected with S protein (25 ng) and a dual-split protein 1-7 (DSP1-7) expression (25 ng) vectors using Lipofectamine 3000 in a 96-well plate. The 293T/ACE2 cells were simultaneously transfected with DSP8-11 (2.5 μg) using Lipofectamine 3000 (Thermo Fisher Scientific, Waltham, MA, USA) in a 6-well plate. Twenty-four hours following transfection, 293T/ACE2 transfected with DSP8-11 were treated with EnduRen (Promega, Madison, WI, USA) at a concentration of 6 μM for 2 h. Subsequently, the cells were co-cultured with 293T cells expressing S and DSP1-7 at a 2:1 ratio for 3 h. The Renilla luciferase activity was quantified using a GloMax luminometer (Promega, Madison, WI, USA). The fusion activity of the S protein was quantified by calculating the luciferase activity relative to that of the wild-type S protein, which was set at 100%.

### 4.8. Immunoprecipitation and Western Blotting

The 293T cells were co-transfected with ACE2-FLAG (1.5 μg) and S protein expression (1.0 μg) vectors using Lipofectamine 3000 in 6-well plate and cultured for 24 h. Cells were lysed in RIPA buffer (50 mM Tris-Cl, pH 7.4, 1% Triton X-100, 0.1% sodium deoxycholate, 0.1% SDS, 150 mM NaCl, and 1 mM EDTA) containing a protease inhibitor cocktail (Nacalai Tesque). The cell lysates were cleared using Protein G Sepharose (Sigma Aldrich, Burlington, MA, USA) for 2 h at 4 °C, and then incubated overnight at 4 °C with anti-FLAG mouse mAb (Wako) or anti-S1 mAb (GeneTex, Irvine, CA, USA). Sepharose-protein G beads were added, and samples were incubated for 1 h at 4 °C. The samples were washed four times with RIPA buffer, resuspended in 2× sample loading buffer, and subjected to sodium dodecyl sulfate–polyacrylamide gel electrophoresis. Proteins were transferred to a polyvinylidene fluoride membrane (Immobilon-P; Millipore, Billerica, MA, USA) and analyzed by Western blotting using anti-S1 (GeneTex, Irvine, CA, USA) or anti-FLAG (Wako) antibodies, followed by staining with anti-mouse (Jackson ImmunoResearch, West Groove, PA, USA) or anti-rabbit IgG conjugated with horseradish peroxidase (HRP; Cell Signaling Technology), and Chemi-Lumi One (Nacalai Tesque). Images were captured using ChemiDoc Touch (Bio-Rad, Hercules, CA, USA) and analyzed using Image Lab Software version 6.1.010.9.0 (Bio-Rad, Hercules, CA, USA).

### 4.9. Laser Scanning Confocal Microscopic Analysis

The 293T cells were seeded on poly L-lysine (Sigma-Aldrich, Burlington, MA, USA)-coated eight-well glass chamber slides (Matsunami Glass, Osaka, Japan) and co-transfected with ACE2-Venus and the S expression vector using Lipofectamine 3000 (Thermo Fisher Scientific, Waltham, MA, USA) according to the manufacturer’s instructions. After 24 h of culture, cells were fixed with 4% paraformaldehyde for 30 min, permeabilized with 0.2% Triton X-100, and stained with anti-S1 antibody. Subsequently, the cells were incubated with anti-rabbit IgG conjugated to Alexa Fluor 555^TM^ (Thermo Fisher Scientific, Waltham, MA, USA), counterstained with DAPI (Sigma-Aldrich, Burlington, MA, USA), and analyzed using an LSM700 confocal laser scanning microscope (Carl Zeiss, Gottingen, Germany) equipped with a 60× objective lens. Image processing was performed using the ZEN microscopy software (ZEN 2009, v5.5.0.375, Carl Zeiss, Gottingen, Germany). For the subcellular localization of ACE2, mCherry-Rab9a (Addgene, Watertown, MA, USA) was co-transfected with ACE2-Venus or an empty vector using Lipofectamine 3000 (Thermo Fisher Scientific, Waltham, MA, USA), and the cells were analyzed as described above.

### 4.10. Statistical Analysis

All statistical analyses were performed using GraphPad Prism software version 10.1.0 (GraphPad Software, Boston, MA, USA).

## 5. Conclusions

In this study, we observed the downregulation of ACE2 by the S protein of SARS-CoV-2. Inhibitors of the protein degradation pathway and confocal microscopic analyses revealed that the S protein interacts with ACE2 in cells, leading to the degradation of ACE2 in the lysosomal compartment via the endocytic pathway. Notably, our findings revealed that the S protein of the Delta variant exhibited a higher ability to downregulate ACE2 than the Omicron variant. This suggests a correlation between the ACE2 downregulation capacity of the S protein and disease severity. Consequently, our study highlights the importance of analyzing the pathogenic characteristics of the SARS-CoV-2 variants based on their ACE2 downregulation activity.

## Figures and Tables

**Figure 1 ijms-25-01353-f001:**
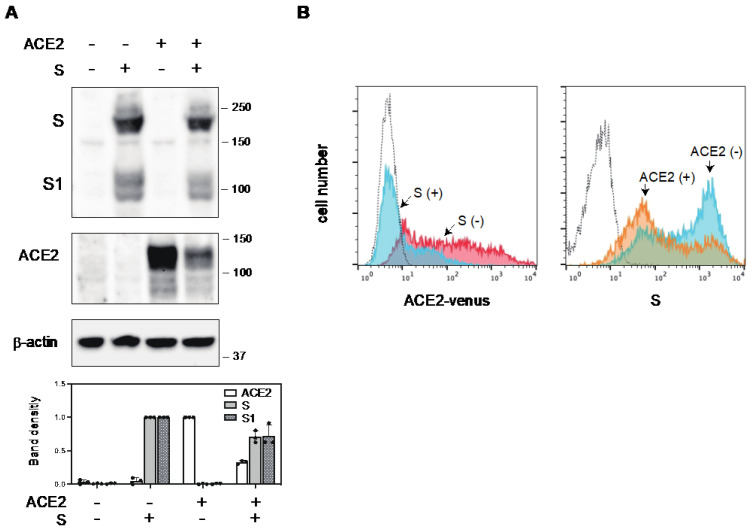
Downregulation of ACE2 by the S protein of SARS-CoV-2. (**A**) Cell lysates from ACE2-expressing 293T cells transfected with either an empty vector or an S protein expression vector were subjected to Western blotting using anti-S1, and-ACE2, and anti-β-actin antibodies. Relative band intensities of ACE2 and the S protein are shown in the lower panel. Band intensities of ACE2 or the S protein without the S protein or ACE2 are shown as 1.0, respectively. (**B**) S protein and ACE2-Venus were co-expressed in 293T cells and stained with anti-S1 antibody. (**Left**) ACE2-Venus expression in the absence (red) or presence (blue) of S protein was analyzed in S1-positive cell population. (**Right**) S1 expression in the absence (blue) or presence (yellow) of ACE2-Venus was analyzed. Dotted line represents the no-antibody-control, non-transfected 293T cells.

**Figure 2 ijms-25-01353-f002:**
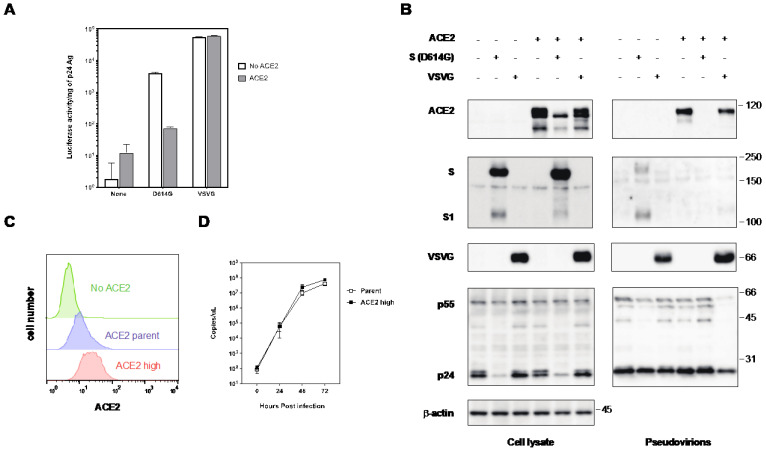
Effect of the over-expression of ACE2 in pseudovirus with S protein and authentic SARS-CoV-2 infection. (**A**) Effect of over-expression of ACE2 in pseudovirus infection with SARS-CoV-2 S protein. HIV-1 pseudoviruses with D614G S protein produced in the absence (white bar) or presence (shaded bar) of FLAG-tagged ACE2 were infected into ACE2-expressing 293T cells. The luciferase activities of infected cells were measured after 48 h of infection in triplicate experiments. (**B**) Western blot analyses of pseudovirus-infected cells and the virions in the absence and presence of FLAG-tagged ACE2. Pseudovirus with S protein or VSVG were produced in 293T cells in the absence or presence of ACE2. Cell lysates and pseudoviruses were analyzed by Western blotting using anti-FLAG, anti-S1, anti-VSVG, anti-ACE2, anti-HIV-1 p24, and anti-β-actin antibodies. To monitor the production of psuedovirions, HIV-1 p55 gag precursor and the final product p24 are shown. The positions of the molecular mass marker (kDa) are indicated on the right. (**C**) The 293T cells (No ACE2), ACE2-expressing parental 293T (ACE2 parent), or highly ACE2-expressing 293T cells (ACE2 high) were stained with an anti-ACE2 antibody and analyzed by flow cytometry. (**D**) ACE2 parent and high-ACE2-293T cells were infected with authentic SARS-CoV-2. Viral RNA titers were determined using real-time PCR in the indicated time in triplicate experiments.

**Figure 3 ijms-25-01353-f003:**
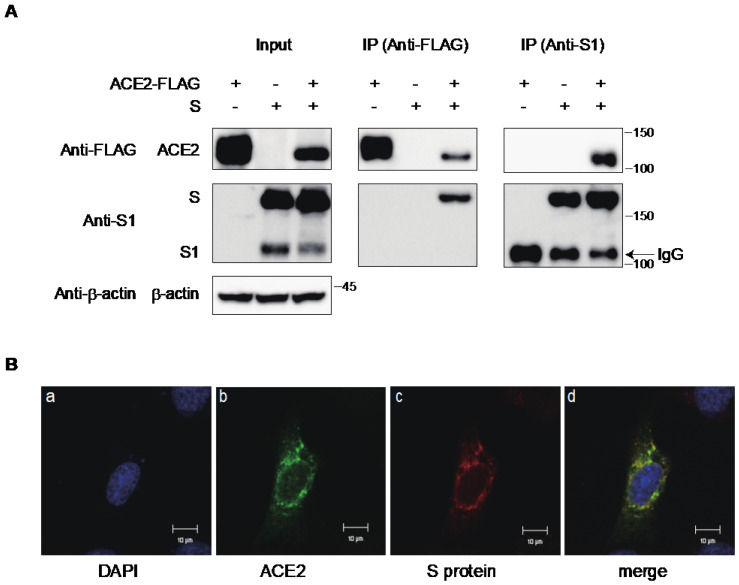
Interaction of ACE2 with the S protein of SARS-CoV-2 in 293T cells. (**A**) Cell lysates from the 293T cells transfected with FLAG-tagged ACE2 and S protein were analyzed by Western blotting using ant-FLAG, anti-S1, anti-β-actin antibodies. Immunoprecipitation (IP) was performed using anti-FLAG and anti-S1 antibodies with protein G-Sepharose beads, and the blots were analyzed with anti-S1 and anti-FLAG antibodies, respectively. The positions of the molecular mass marker (kDa) are indicated on the right. (**B**) Confocal microscopic images of the 293T cells transfected with ACE2-Venus (green) and the S protein. Panels show the counterstaining of cell nuclei with DAPI (**a**), ACE2-Venus (**b**), S protein (**c**), and merged image (**d**). Scale bars correspond to 10 μm.

**Figure 4 ijms-25-01353-f004:**
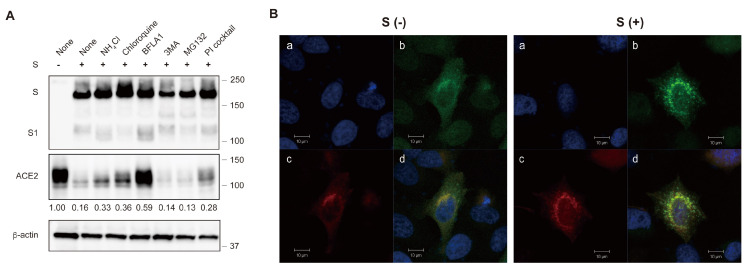
The degradation of ACE2 by the S protein in the lysosomal compartment via the endocytic pathway. (**A**) ACE2-expressing 293T cells were transfected with the S protein in the presence of various inhibitors targeting protein degradation pathways. Cell lysates were analyzed by Western blotting using anti-S1, anti-ACE2, and anti-β-actin antibodies. The positions of the molecular mass marker (kDa) are indicated on the right. The relative band intensities of ACE2 are shown on the bottom of ACE2 image. The band intensity of ACE2 without the S protein is shown as 1.0. (**B**) Confocal microscopic images of 293T cells transfected with ACE2-Venus and Rab9a-mCherry. The panels show counterstaining with cell nuclei with DAPI (**a**), ACE2-Venus (**b**), Rab9a-mCherry (**c**), and a merged image (**d**) in the absence (**left**) and presence (**right**) of the S protein. Scale bars correspond to 10 μm.

**Figure 5 ijms-25-01353-f005:**
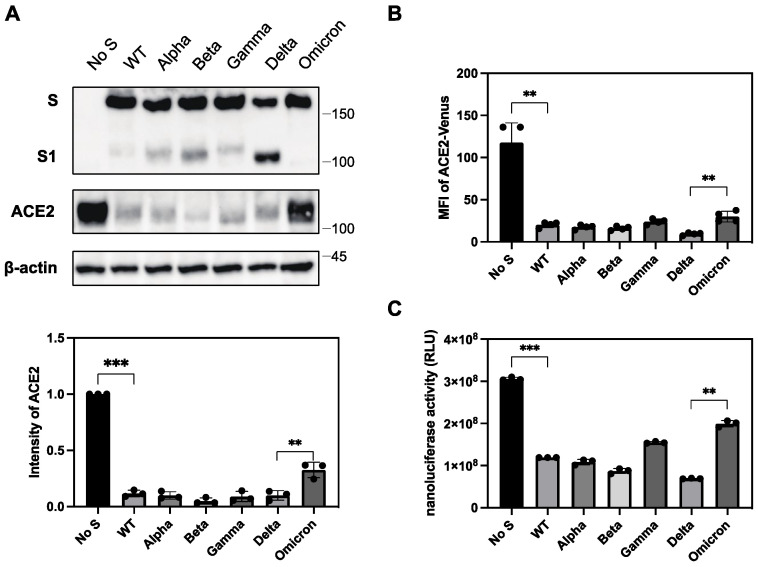
The differential ability of the S protein from various VOCs to downregulate ACE2. (**A**) (**upper**) Cell lysates from ACE2-expressing 293T cells transfected with S protein from previous VOCs were analyzed by Western blotting using ant-ACE2, anti-S1, and anti-β-actin antibodies. The positions of the molecular mass marker (kDa) are indicated on the right. (**lower**) The relative band intensities of ACE2 in the presence of the S protein from previous VOCs in Western blotting were analyzed and compared using Image Lab software (BioRad) in triplicate experiments. The band intensity of ACE2 without the S protein is shown as 1.0. Bars depicts means and standard deviation (** *p* < 0.01, *** *p* < 0.001 using unpaired *t*-test). (**B**) The S protein and ACE2-Venus were co-expressed in 293T cells and stained with anti-RBD polyclonal antibody. The ACE2 downregulation activity of the S protein of previous VOCs was determined using the mean fluorescence intensity (MFI) of ACE-Venus protein in the S-protein-positive cell population. The column and bar indicate means and standard deviation in triplicate experiments (** *p* < 0.01 using unpaired *t*-test). (**C**) S protein and ACE-nLuc were co-expressed in 293T cells in the presence or absence of the S protein from previous VOCs. The ACE2 downregulation activity of the S protein from previous VOCs was calculated using the nanoluciferase activity of the wild-type S proteins as 100% (** *p* < 0.01, *** *p* < 0.001 using unpaired *t*-test).

**Figure 6 ijms-25-01353-f006:**
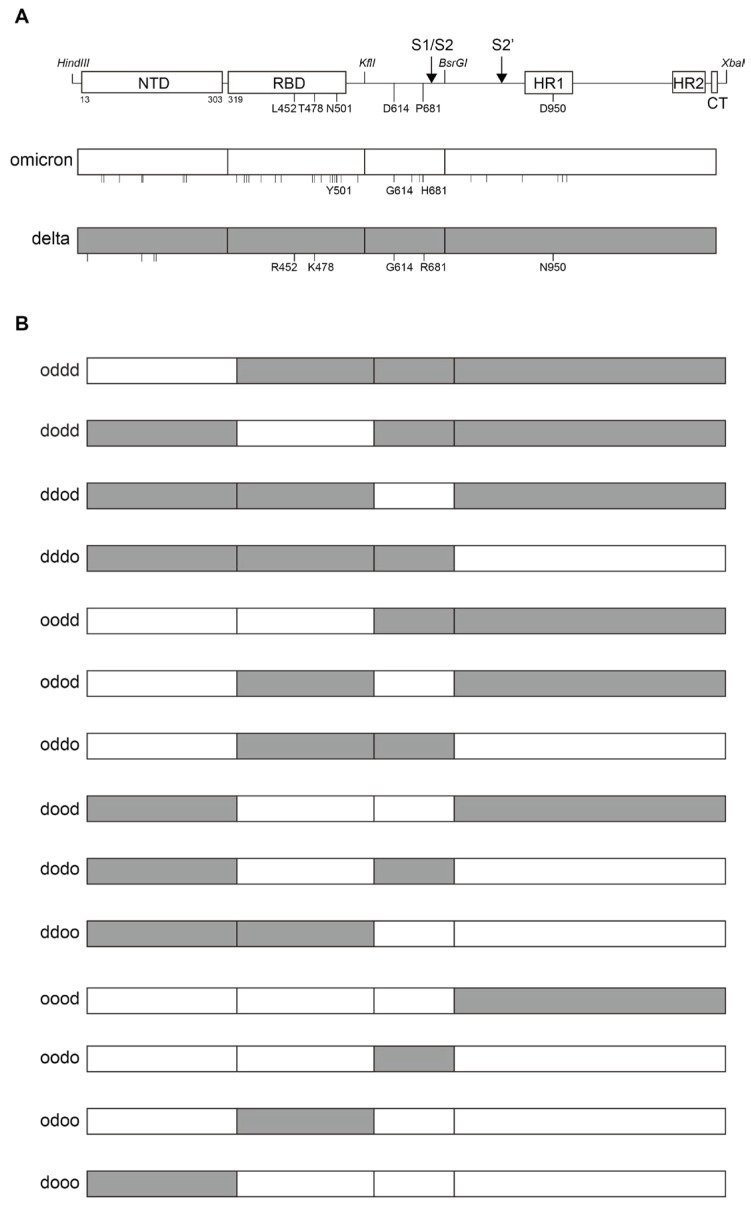
Structures of the S protein and chimeras between Omicron and Delta variants. (**A**) Schematic representation of the S protein structure of SARS-CoV-2 with shared restriction enzyme sites and its variants, Omicron and Delta. Only the representative mutations in the Omicron and Delta variants are shown. Other mutations are depicted in vertical bars under each S protein. (**B**) The structures of the chimeric S protein between Omicron and Delta variants. White and grey colors represent the S protein from the Omicron and Delta variants, respectively.

**Table 1 ijms-25-01353-t001:** ACE2 downregulation and fusion activities of chimeras between the S protein from the Omicron and Delta variants.

	ACE2 Downregulation Activity (%) ^a^	Fusion Activity (%) ^b^
S wild	100.0 ± 0.4 ^c^	100.0 ± 8.2
Delta	142.3 ± 0.5	257.7 ± 6.3
Omicron	75.4 ± 3.5	13.3 ± 1.6
oddd	134.8 ± 1.1	212.8 ± 8.6
dodd	93.5 ± 1.9	176.8 ± 5.8
ddod	113.3 ± 1.2	95.6 ± 4.8
dddo	110.2 ± 2.7	127.9 ± 5.2
oodd	111.7 ± 3.3	184.9 ± 9.0
odod	104.1 ± 2.4	71.1 ± 5.8
oddo	107.0 ± 3.7	117.3 ± 3.9
dood	104.6 ± 2.2	83.7 ± 9.8
dodo	47.8 ± 3.9	48.4 ± 2.6
ddoo	88.5 ± 3.1	26.2 ± 0.7
oood	117.4 ± 3.0	67.5 ± 3.7
oodo	67.7 ± 1.2	46.7 ± 4.7
odoo	77.5 ± 2.0	24.5 ± 3.1
dooo	75.9 ± 2.2	13.4 ± 1.1

^a^ The ACE2 downregulation activity of each chimera was determined using the 293T cells carrying nanoluciferase-linked ACE2. The reduction in nanoluciferase activity caused by the introduction of the S protein was determined and compared to the use of an empty vector as a control. The ACE2 downregulation activity of each chimera was calculated using the nanoluciferase activity of the wild-type S proteins as 100%. ^b^ The fusion activity of the S protein chimera was determined by a DSP-based cell fusion assay using 293T cells expressing the S protein and DSP1-7, co-cultured with the 293T cells expressing ACE2 and DSP8-11. Renilla luciferase activity was measured after 3 h of coculture. The fusion activity of each chimera was determined using the Renilla luciferase activity of the S protein as 100%. ^c^ Mean ± SD (*n* = 4).

**Table 2 ijms-25-01353-t002:** ACE2 downregulation and fusion activities of Delta, Omicron, and wild-type S with mutations.

	ACE2 Downregulation Activity (%) ^a^	Fusion Activity (%) ^b^
S wild (Wuhan)	100.0 ± 5.9 ^c^	100.0 ± 5.7 ^c^
Delta	156.4 ± 4.8	196.5 ± 18.3
Omicron	71.0 ± 7.7	14.3 ± 2.6
L452R	107.6 ± 5.5	97.0 ± 9.3
P681R	95.3 ± 11.4	166.5 ± 12.1
D950N	117.1 ± 4.4	124.2 ± 10.2
L452R/P681R	120.6 ± 6.4	170.8 ± 12.9
L452R/D950N	111.8 ± 5.7	122.2 ± 5.2
P681R/D950N	133.5 ± 4.8	196.6 ± 10.7
L452R/P681R/D950N	156.6 ± 5.9	187.5 ± 6.8

^a^ The ACE2 downregulation activities of the S protein mutants were determined using 293T cells carrying nanoluciferase-linked ACE2. The reduction in nano-luciferase activity via the introduction of S protein mutants was determined and compared to that of the empty vector, used as a control. The ACE2 downregulation activities of the S protein mutants were calculated, using the ACE2 downregulation activity of wild-type S proteins as 100%. ^b^ The fusion activities of S protein mutants were determined by a DSP-based cell fusion assay, using 293T cells expressing the S protein, and DSP1-7, co-cultured with 293T cells expressing ACE2 and DSP8-11. Renilla luciferase activity was measured after 3 h of coculture. The fusion activities of the S mutants were calculated using the Renilla–luciferase activity of the S protein as 100%. ^c^ Mean ± SD (*n* = 4).

## Data Availability

All data associated with this study are presented in the paper or Appendix A. All the raw data are available upon request.

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
