# Peer review of "Differential Ability of Spike Protein of SARS-CoV-2 Variants to Downregulate ACE2"

_ijms, 2024, doi:10.3390/ijms25021353_

Round 1
Reviewer 1 Report
Comments and Suggestions for Authors
In this manuscript entitled “Differential ability of S protein of SARS-CoV-2 variants to downregulate ACE2“, the authors Maeda et al, have hypothesized that S protein of the SARS-CoV-2 is able to downregulates the ACE2 expression in infected cells. To test this they used ACE2 expressing HEK293T cells. Such studies have already been performed previously and in fact using appropriate and better cell lines and models such as primary lung alveolar cells (PMID: 34149696) which express ACE2 endogenously. Furthermore the authors claim that this downregulation is occurring through lysosomal degradation by showing co-localization with Rab9 in the cytoplasm. Again, this has already been established in the scientific community (PMID: 36287912) using in vitro as well as in vivo data from HEK and hamster tissue respectively. The experimental strategy used in this paper seems to be a duplicated/repeated version of previously published papers in a more artificial setup of “ACE2 expressing HEK cells” which in my opinion significantly reduces the novelty and the value of the data presented here.
The methods used in this study are standard but several points/statements are unclear which raises concerns regarding quality and robustness of the data shown in the manuscript. Also as mentioned before most of the conclusions made/shown by authors have already been previously published and so I feel a serious lack of novelty in this presented work.
· What do the authors means by ‘authentic’ SARS-CoV-2 ? This word has been used several times in the text.
· No quantification of western blots were provided (except for Fig 5A). please include analysis.
· Typo- line 39- “omicorns” instead of ‘omicron’. The ‘s’ should be removed.
· Line 49- the authors write “previous data have” but then do not provide any reference.
There are several parts in the text where statements/data is mentioned but without proper reference /citation for the same. For eg, in line 64-65, line Please improve
· Fig 1B- Both ACE2 and S expression seem to affect/modulate each other. Can the authors explain why there is a difference in S expression whether ACE2 is present/absent as shown in flow cytometry data.
· Line 90- not clear what the authors mean by “293T cells in pseudovirus-producing cells”
· Line 109- there is no Figure 2D provided in the figure panel. Also authors should add somewhere in the text that what are p55 and p24 proteins and why they tested for them. Abbreviate VLP.
· The authors initially hypothesize about fusion activity of S protein is resulting in ACE2 downregulation (in line 241-243) but later on in line 296 (and line 377)- they again go against their original hypothesis and do not even provide any explanation for this. What is the main message here and what do the authors aim to convey? What do they mean by ‘other virological mechanisms’ (in line 377)?
· Line 329-330: the authors write ‘endogenous’ ACE2- I would argue that this is incorrect. The authors artificially /exogenously overexpressed ACE2 in HEK293T cells to perform their experiments. HEK293T is not a cell line that produces high levels of ACE2 endogenously. This should be repharsed. Also the authors mentioned (line 454) that they co-transfected ACE2 and S in HEK cells but did not provide the transfection protocol/ details for the experiments.
· Line 428- why do the authors mention pseudotyping of HIV-1 infection? I thought the experiments were performed for SARS-CoV-2 S pseudoviruses. Please clarify.
Author Response
Response to Reviewer 1 Comments
- Summary
Thank you very much for taking the time to review our manuscript. Please find the detailed responses below and the corresponding revisions/corrections highlighted/in track changes in the re-submitted files.
- Response and Revisions
We thank the reviewer’s comments. These comments were helpful, and gave us a better perspective of our work. The point-by-point comments were as bellows.
- Point-by-point response to Comments
In this manuscript entitled “Differential ability of S protein of SARS-CoV-2 variants to downregulate ACE2“, the authors Maeda et al, have hypothesized that S protein of the SARS-CoV-2 is able to downregulates the ACE2 expression in infected cells. To test this they used ACE2 expressing HEK293T cells. Such studies have already been performed previously and in fact using appropriate and better cell lines and models such as primary lung alveolar cells (PMID: 34149696) which express ACE2 endogenously.
Response: We appreciate the reviewer’s comments. As the reviewer suggested, several data were similar to previous works, and our data were always derived from HEK293T cells not from the primary lung cells. However, the previous work showing the ACE2 downregulation by the S protein was determined by transcriptional level not the protein level (PMID: 34149696). Instead, we were able to confirm that the interaction of the S protein with ACE2 in the same cells directly induces the downregulation of ACE2. The analyses of co-immunoprecipitation and confocal microscopy show the degradation of ACE2 is not through the transcriptional regulation of ACE2 by the S protein. This kind of analysis was only possible to use the artificial setting such as using ACE2 highly expressing HEK293T cells. Therefore, we revised our manuscript showing the downregulation of ACE2 by the interaction of the S protein at the protein level at lines 330-335 in the discussion.
Furthermore the authors claim that this downregulation is occurring through lysosomal degradation by showing co-localization with Rab9 in the cytoplasm. Again, this has already been established in the scientific community (PMID: 36287912) using as well as data from HEK and hamster tissue respectively. The experimental strategy used in this paper seems to be a duplicated/repeated version of previously published papers in a more artificial setup of “ACE2 expressing HEK cells” which in my opinion significantly reduces the novelty and the value of the data presented here.
Response: Thank you for the reviewer’s comments. Previous work (PMID: 36287912) showed that the addition of soluble form of the S protein into ACE2-expressing cells leaded to the ACE2 downregulation in lysosomal compartment. However, we wanted to check how ACE2 molecules in infected cells are processed by de novo synthesized S protein in the same cells. To this end, we established the ACE2-expressing cells, then introduced the S protein similar to the infected cells. We think our findings show the degradation of ACE2 in lysosomal compartment in the expression of the S in cis not trans. We therefore revised our manuscript showing that de novo synthesized S protein induced ACE2 degradation in lysosomal compartment at lines 345-347 in the discussion.
The methods used in this study are standard but several points/statements are unclear which raises concerns regarding quality and robustness of the data shown in the manuscript. Also as mentioned before most of the conclusions made/shown by authors have already been previously published and so I feel a serious lack of novelty in this presented work.
Response: Although the reviewer suggested that some of our results were similar and not novel, our present data were somewhat different from previous works as mentioned above. In addition, our new findings were differential ability of ACE2 downregulation of S protein of variant to downregulate ACE2 as shown in the title. We further determined the responsible region of ACE2 downregulation by the S protein using mutants and chimeras between the S protein of Delta and Omicron, which has not been previously described elsewhere.
What do the authors means by ‘authentic’ SARS-CoV-2 ? This word has been used several times in the text.
Response: We used two assay systems for SARS-CoV-2 infection. One is HIV-based pseudotype SARS-CoV-2 S which ensures the single-round infection and check the entry efficiency by SARS-CoV-2 S protein. Another one is intact SARS-CoV-2 infection to monitor the replication fitness in the multiple-round of infection. To distinguish them, we used the term “authentic” in usual SARS-CoV-2 infection.
- No quantification of western blots were provided (except for Fig 5A). please include analysis.
Response: We have added quantified data of western blot in Fig. 1A in revised manuscript. In Fig. 4, only the band intensities of ACE2 have been shown on the bottom of ACE2 image of western blot. We do not think that the showing other the quantified data of western blot in Fig. 2 and Fig. 3 is meaningful. We have revised the legends of Fig. 1 and Fig. 4 accordingly at lines 85-86, and 169-170, respectively.
- Typo- line 39- “omicorns” instead of ‘omicron’. The ‘s’ should be removed.
Response: We have to apologize the typological mistake, which has been corrected at line 39 in the introduction.
- Line 49- the authors write “previous data have” but then do not provide any reference.
Response: We have to apologize the missing of the references. We have incorporated the references at lines 49-50 in the introduction.
There are several parts in the text where statements/data is mentioned but without proper reference /citation for the same. For eg, in line 64-65, line Please improve
Response: We have to apologize the missing of the references. We have added the references at lines 63-66. Several references were also incorporated accordingly.
Fig 1B- Both ACE2 and S expression seem to affect/modulate each other. Can the authors explain why there is a difference in S expression whether ACE2 is present/absent as shown in flow cytometry data.
Response: Flow cytometric data shows the surface expression levels of the proteins while western blot data shows the total expression levels in the cells. Therefore, it was speculated that the surface expression of the S protein was inhibited by ACE2 expression. We added these sentences in the manuscript at lines 77-78, and lines 335-339.
Line 90- not clear what the authors mean by “293T cells in pseudovirus-producing cells”
Response: We are sorry for the confusing expression. We wanted to analyze the effect of ACE2 expression in infected cells on the infectivity of nascent SARS-CoV-2 S pseudovirions. The sentence has been modified accordingly at lines 92-95.
Line 109- there is no Figure 2D provided in the figure panel.
Response: We have to apologize the missing of Fig. 2D. That figure was actually shown Fig. 2 in original manuscript, but “D” was missing. Therefore, Fig. 2 has been replaced showing Fig. 2D.
Also authors should add somewhere in the text that what are p55 and p24 proteins and why they tested for them. Abbreviate VLP.
Response: We have to apologize to miss the description of p55 and p24: p55 is the percussor of HIV-1 gag protein while p24 is the final gag product of HIV-1. The analyses of these bands in western blot were necessary to show the pseudovirions were properly generated. These expressions have been incorporated at lines 123-124 in Fig. 2 legend. The term “VLP, virus-like particle”, has been deleted and changed to “pseudovirions” at line 123. We think the use of “pseudovirions” is better in this case.
- The authors initially hypothesize about fusion activity of S protein is resulting in ACE2 downregulation (in line 241-243) but later on in line 296 (and line 377)- they again go against their original hypothesis and do not even provide any explanation for this.
Response: We are sorry for the confusion. We first hypothesized the fusion activity of the S protein is correlated with ACE3 downregulation activity since the S protein of Delta variant had highest fusion and ACE2 downregulation activity while that of Omicron variant exhibited the lowest fusion and ACE2 downregulation activity. However, the chimeric studies between the S protein of Delta and Omicron showed that the fusion and ACE2 downregulation activity of S protein chimera were not simply correlated. Therefore, we revised our expression at lines 253-259 and 383-389 to avoid the confusion.
What is the main message here and what do the authors aim to convey? What do they mean by ‘other virological mechanisms’ (in line 377)?
Response: We appreciate the reviewer’s comment. We added possible mechanisms such as binding affinity to ACE2, or the proteolytic cleavage efficiency of the S protein to explain several inconsistencies between fusion and ACE downregulation activity of the chimeric S protein at lines 395-398.
- Line 329-330: the authors write ‘endogenous’ ACE2- I would argue that this is incorrect. The authors artificially /exogenously overexpressed ACE2 in HEK293T cells to perform their experiments. HEK293T is not a cell line that produces high levels of ACE2 endogenously. This should be repharsed.
Response: We have to apologize the mistake to use the term “endogenous”. As we mentioned above, we wanted to check the effect of de novo synthesized S protein and ACE2 in the same cells. Therefore, the sentence has been changed to use the term “de novo” instead of the use “endogenous” at lines 345-347.
Also the authors mentioned (line 454) that they co-transfected ACE2 and S in HEK cells but did not provide the transfection protocol/ details for the experiments.
Response: The method for transfection has been revised in the materials and methods accordingly.
Line 428- why do the authors mention pseudotyping of HIV-1 infection? I thought the experiments were performed for SARS-CoV-2 S pseudoviruses. Please clarify.
Response: We have to apologize the confusion. As we mentioned above, we used HIV-based pseudotyped SARS-CoV-2 S (reviewed in PMID32692348) for assessing the impact of ACE2 expression in the infectivity of the nascent pseudovirions.
Reviewer 2 Report
Comments and Suggestions for Authors
Thank you for your fine article. I would like to comment on the following 3 points.
1. I was concerned about the omission of references in the introduction. Please include the references in the introduction (p. 1, l. 28-41 and p. 2, l. 46-50).
2. Can the authors explain the results of the alpha, beta, and gamma variants? Also, is there no significant difference between gamma and delta in Figures 5B and 5C?
3. It is unclear how the results of this study will be used in clinical practice. Please mention this if the authors wish.
Author Response
Response to Reviewer 2 Comments
- Summary
Thank you very much for taking the time to review our manuscript. Please find the detailed responses below and the corresponding revisions/corrections highlighted/in track changes in the re-submitted files.
- Response and Revisions
We thank the reviewer’s comments. These comments were helpful, and gave us a better perspective of our work. The point-by-point comments were as bellows.
- Point-by-point response to Comments
- I was concerned about the omission of references in the introduction. Please include the references in the introduction (p. 1, l. 28-41 and p. 2, l. 46-50).
Response 1: We appreciate the reviewer’s comment. We have included several references at lines 28-50 in the introduction.
- Can the authors explain the results of the alpha, beta, and gamma variants? Also, is there no significant difference between gamma and delta in Figures 5B and 5C?
Response 2: As the reviewer suggested, there were a statistically significant differences between gamma and delta variants in Figure 5B and 5C. However, these was no significance in Figure 5A. In contrast, statistically significant difference between delta and omicron was consistently observed in all assays. Therefore, we have partially incorporated these descriptions at lines 198-202, 210-212, and 364-368.
- It is unclear how the results of this study will be used in clinical practice. Please mention this if the authors wish.
Response 3: We appreciate the reviewer’s comment. As we discussed at lines 402-406 in the end of discussion, we hope our strategy may help the prediction of pathogenic characteristics of newly emerged variant in future.
Round 2
Reviewer 2 Report
Comments and Suggestions for Authors
Thank you for your response. There are no additional comments. Good luck!